# Abstinence Restores Cardiac Function in Mice with Established Alcohol-Induced Cardiomyopathy

**DOI:** 10.3390/cells12242783

**Published:** 2023-12-07

**Authors:** Joshua M. Edavettal, Nicholas R. Harris, Sarah E. Cohen, Janos Paloczi, Bysani Chandrasekar, Jason D. Gardner

**Affiliations:** 1Department of Physiology, LSU Health Sciences Center—New Orleans, New Orleans, LA 70112, USA; jedav1@lsuhsc.edu (J.M.E.); nharr8@lsuhsc.edu (N.R.H.); scohe2@lsuhsc.edu (S.E.C.); jpaloc@lsuhsc.edu (J.P.); 2Department of Medicine, University of Missouri School of Medicine, Columbia, MO 65212, USA; chandrasekarb@health.missouri.edu; 3Department of Medical Pharmacology and Physiology, Dalton Cardiovascular Research Center, and Harry S. Truman Memorial Veterans Hospital, Columbia, MO 65212, USA

**Keywords:** alcohol-induced cardiomyopathy, abstinence, reversal, cardiac function

## Abstract

Alcohol-induced cardiomyopathy (ACM) has a poor prognosis with up to a 50% chance of death within four years of diagnosis. There are limited studies investigating the potential of abstinence for promoting repair after alcohol-induced cardiac damage, particularly in a controlled preclinical study design. Here, we developed an exposure protocol that led to significant decreases in cardiac function in C57BL6/J mice within 30 days; dP/dt max decreased in the mice fed alcohol for 30 days (8054 ± 664.5 mmHg/s compared to control mice: 11,188 ± 724.2 mmHg/s, *p* < 0.01), and the dP/dt min decreased, as well (−7711 ± 561 mmHg/s compared to control mice: −10,147 ± 448.2 mmHg/s, *p* < 0.01). Quantitative PCR was used to investigate inflammatory and fibrotic biomarkers, while histology was used to depict overt changes in cardiac fibrosis. We observed a complete recovery of function after abstinence (dP/dt max increased from 8054 ± 664 mmHg/s at 30 days to 11,967 ± 449 mmHg/s after abstinence, *p* < 0.01); further, both inflammatory and fibrotic biomarkers decreased after abstinence. These results lay the groundwork for future investigation of the molecular mechanisms underlying recovery from alcohol-induced damage in the heart.

## 1. Introduction

Alcohol consumption is a pervasive risk factor across myriad pathologies, with 60% of women and 80% of men in developed countries having consumed alcohol at least once in their lifetime [1]. Over 200 million people in the United States have consumed alcohol at least once, and over 170 million have consumed alcohol in the past year, while 16 million people have had heavy alcohol consumption in the past month [2]. The risk of developing an alcohol use disorder (AUD) in the last year was approximately 10% [1]. Furthermore, over 2.4 million deaths across the world are attributed to alcohol use [3], making alcohol use the eighth leading risk factor for deaths globally [4].

Although the scope of this study was to investigate the effects of alcohol on the heart, alcohol and its metabolites detrimentally affect multiple organs and organ systems. Previous work has characterized the damage alcohol does to the body [5,6,7,8], but despite the risk alcohol poses to the cardiovascular system, there is a lack of research on alcohol-induced cardiomyopathy (ACM), particularly in preclinical models aimed at intervention. 

The heart is at risk of alcohol-induced damage, both in acute and chronic consumption. Many studies have investigated the hormetic “J-curve” association between various cardiovascular pathologies and alcohol consumption, but there are studies that have shown that even in acute settings, alcohol consumption can be damaging to the heart. The risk of arrhythmias, particularly atrial fibrillation, increases after as little as one drink per day [9]. High doses of alcohol increase heart rate and blood pressure 13 h after consumption [10], and there is a linear association between systolic blood pressure and any amount of alcohol consumption. Further, in the first 24 h after alcohol consumption, there is an increased risk of myocardial infarction and hemorrhagic and ischemic stroke [11]. Chronic alcohol consumption has been linked to other pathological cardiovascular states. AUD can lead to ischemic heart disease, hypertension [12,13], cerebrovascular events [14], persistent arrhythmias (atrial fibrillation being the most common [15]), both systolic and diastolic dysfunction [16,17], cardiomyopathy, and heart failure [18]. ACM is the leading cause of non-ischemic dilated cardiomyopathy [3], marked by significant fibrosis and both diastolic and systolic dysfunction [19], eventually leading to heart failure. Current treatment protocols rely on treating the symptoms of heart failure and the patient ceasing alcohol use [20]. Much of the current preclinical research on alcohol and ACM focuses on the mechanisms of alcohol-induced damage [21], with very limited research carried out to characterize the effects of abstinence. There has been some research on AUD recovery and alcohol abstinence in human populations, though much of this is focused on the neurocognitive and psychological aspects of addiction and treatment. 

There is significant potential to recover from alcohol-induced damage with adherence to abstinence. Thomes recently published a review on the regenerative capacity of the human body to repair and overcome alcohol damage after abstinence; while this review focuses primarily on liver and GI tract recovery, evidence of bone and heart recovery is also discussed [22]. Interestingly, there is potential to recover function even after years of heavy alcohol use. One case study examined a patient with systolic heart failure, who presented with delirium tremens (severe alcohol withdrawal) and an ejection fraction (EF) of 20% with elevated cardiac troponin T. Abstinence from alcohol for one month led to an increase in EF to above 60% (a typical EF in a healthy person) [23]. A recent study used echocardiography to assess morphological and functional changes in the hearts of active heavy-drinking patients; after 6 months of abstinence mean E/e’ ratio improved and posterior wall thickness increased [24]. However, other than a few accounts of post-abstinence recovery in humans, there are limited data on cardiac recovery from alcohol-induced dysfunction. To our knowledge, this recovery process has not been studied in animal models of ACM.

Here, using a mouse model of ACM, we assess the effects of 30 days of abstinence following established cardiac dysfunction. We performed echocardiography to monitor changes in left ventricular wall dimensions, left ventricular catheterization to assess cardiac systolic and diastolic function, histological staining and quantification of collagen, and analysis of mRNA expression of both fibrotic and inflammatory markers.

## 2. Materials and Methods

### 2.1. Study Design and Timeline

Mice were acclimated to the control liquid diet for five days before being randomly assigned to 2 groups. The experimental group received a 5% ethanol liquid diet and the control group received an isocaloric diet without ethanol. Both groups received a binge gavage dose at days 10 and 30 (5 g/kg dose of a 31.5% ethanol solution and a 9 g/kg dose of a maltose–dextrin solution). At least 12 h after the gavage, echocardiography was performed to assess morphologic changes. After day 30, a subset of each group was euthanized for terminal cardiac functional measures and tissue collection. Remaining mice in the experimental group were switched from the alcohol diet back to the control diet for an additional 30 days (these were the abstinence mice), while the control group continued to receive the control diet. On day 60, echocardiography was performed again. Terminal left ventricular catheterization was performed the subsequent day to assess functional changes (Figure 1). 

### 2.2. Animals

Adult, male C57BL6/J mice were obtained from the Jackson Laboratory (Bar Harbor, ME, USA) at 9–10 weeks old. Mice were given a week to acclimate to the animal care facilities at Louisiana State University Health Sciences Center (LSUHSC), before starting the liquid diet; these facilities are humidity and temperature controlled, and animals were monitored daily. The temperature was kept above 23 °C to prevent hypothermia, as alcohol causes the core body temperature to fall. Before the liquid diet was given, animals were fed a standard chow diet (Teklab Extruded Rodent Diet 2019S) and had ad libitum access to water. Mice were kept on a traditional 12 h/12 h light–dark cycle (6:00 a.m.–6:00 p.m. in light). All experiments and procedures were approved by the LSUHSC Institutional Animal Care and Use Committee (Protocol No. 3859). 

### 2.3. Liquid Diet

The liquid diet was purchased from Bio-Serv, Flemington, NJ, USA (Products: F1258SP (Ethanol Shake and Pour Diet) and F1259SP (Control Shake and Pour Diet)) and administered in feeder tubes also purchased from Bio-Serv (Product Nos. 9019 and 9015). The control and ethanol diets were isocaloric diets, with ethanol accounting for 35% of calories in the ethanol diet. The diets were prepared as outlined in Bertola et al.; this chronic + binge model produces blood alcohol levels of ~180 mg/dL after 10 days on the 5% ethanol liquid diet and ~400 mg/dL after ethanol binges [25]. Administration of the diet followed a modified version of the chronic + binge model. In our diet protocol, mice were given 5 days to acclimate to the liquid diet, where all animals were given the control liquid diet. After this period of acclimation, mice were randomly split into experimental and control groups. Experimental groups received a 5% ethanol liquid diet, while control animals continued receiving the control diet. Mice were pair-fed. The diet was administered fresh daily, within 2 h of the dark cycle. The ethanol diet was continued for 30 days to induce cardiomyopathy; then, a subgroup of mice was switched to the control diet (abstinence group) for an additional 30 days (~65 days total on liquid diets). 

### 2.4. Gavage

Binge dosing by gavage was performed on days 10 and 30. Gavage was administered in the light portion of the light–dark cycle. Control mice received a maltose–dextrin solution (9 g/kg of body weight), while ethanol mice received a 31.5% (*v*/*v*) ethanol solution (5 g/kg of body weight). After gavage, mice were placed on a heated pad for an hour and then returned to their original cage location in the animal housing facility.

### 2.5. Echocardiography

To examine differences in left ventricular wall dimensions, echocardiography was performed after day 30 and day 60. Echocardiography was started a minimum of 12 h after ethanol gavage to avoid any cardio-depressive effects attributed to acute alcohol intoxication. A Vevo 3100 Imaging System and a 30 MHz probe from VisualSonics (Toronto, ON, Canada) were used. Isoflurane (1–2%) was used to immobilize the mice for imaging. Mice were kept on a 37 °C heated pad throughout the duration of the recordings, and the total time under anesthesia and obtaining measurements was under 15 min. Short and long axis views were obtained in B-mode and M-mode. Analysis was performed on M-mode images, using VisualSonics software (Vevo Lab 5.7.1). All measurements were made over a minimum of three cardiac cycles, and the averages were used for further analysis.

### 2.6. Left Ventricular Catheterization

After echocardiographic measurements were obtained, mice underwent terminal open chest left ventricular (LV) pressure-volume catheterization. LV catheterization was performed to assess systolic and diastolic function. Isoflurane at 3–4% was used for induction before mice were intubated; after intubation, anesthesia was maintained with 2–3% isoflurane. Mice were kept on a heated pad throughout the procedure. First, the skin on the anterior of the chest was dissected, revealing the abdominal muscles and the linea alba. An incision was made along the linea alba, starting inferior to the zyphoid process and through the sternum. Bleeding was minimal and controlled using both cauterization and bulldog clips as needed. After the pericardium was transected, the heart was revealed, and a 25 G needle was used to puncture the apex of the LV. A pressure-volume transducer (SPR-839; Millar, Houston, TX, USA) was inserted into the LV, through the hole created by the needle. The positioning of the transducer was achieved via observation of the pressure-volume loops that were plotted in real time. Anesthesia was closely monitored, and a heat lamp was used to aid core body temperature. For steady-state measurements, the transducer was kept in the left ventricle over several cardiac cycles. For load-independent measures, occlusion of the inferior vena cava was performed using blunted forceps covered with silastic tubing. Occlusion of the IVC reduces venous return (preload), which decreases the filling volume of the LV. The resulting PV loops are used to determine load-independent measures of cardiac function. E(es) is the slope of the end-systolic pressure-volume relationship, while preload-recruitable stroke work (PRSW) is the slope of stroke work versus end-diastolic volume. Both are measures of load-independent contractility. After occlusion data were collected, approximately 0.1 mL of 3% hypertonic saline was injected into the jugular vein for determination of parallel conductance. Volume calibration was performed using blood-filled cuvettes (Millar, Product No. 910–1049). LV pressures and volumes were recorded and analyzed using Labchart 8 cardiac axis pressure-volume loop software (Ver. 8.1.24).

### 2.7. Heart Collection

After catheterization and recordings were obtained, hearts were removed and placed in cold saline. They were gently blotted with gauze, and the total heart weight was recorded. Under a microscope, hearts were dissected. Atria were removed, and the right ventricle (RV) was separated from the LV and interventricular septum (IVS). RV weights were recorded, as were LV and septal weights. Then, the IVS was removed from the LV. The RVs and LVs with separated IVSs were snap frozen in cryotubes using liquid nitrogen. Tibias were obtained for normalization of weights to tibial lengths. 

### 2.8. Quantitative Real-Time PCR (qPCR)

LV samples were homogenized in Fisherbrand^TM^ Pre-Filled Bead Mill Tubes (1.4 mm ceramic beads, Cat. # 15-340-153) using Benchmark Scientific BeadBlaster^TM^. Total RNA was isolated using an RNeasy Lipid Tissue Mini Kit (Qiagen, Hilden, Germany, Cat. # 74804) according to the manufacturer’s protocol. RNA concentration and quality were analyzed using a NanoDrop^TM^ 1000 (Thermo Scientific, Waltham, MA, USA). RNA (2 µg) was reverse transcribed with the High-Capacity cDNA Reverse Transcription Kit (Applied Biosystems^TM^, Foster City, CA, USA, Cat. # 4368814) according to the manufacturer’s protocol with the following cycling conditions: 25 °C for 10 min, 37 °C for 120 min, and 85 °C for 5 min. qPCR was performed using TaqMan Fast Advanced Master Mix (Thermo Fisher Scientific, Cat. # 4444963) in a CFX Opus 96 Real-Time PCR System (BioRad, Hercules, CA, USA) for the following targets: interleukin 1 beta (IL-1β, assay Mm00434228_m1), interleukin 6 (IL-6, assay Mm00446191_m1), interleukin 10 (IL-10, assay Mm01288386_m1), La Ribonucleoprotein 6 (LARP6, assay Mm00470891_m1), collagen type 1 alpha-1 (COL1A1, assay Mm00801666_g1), and collagen type 3 alpha-1 (COL3A1, assay Mm00802300_m1). The cycling conditions for qPCR included polymerase activation (95 °C for 20 s) followed by 40 cycles of denaturing (95 °C for 1 s) and annealing/extending (60 °C for 20 s). Relative quantification was calculated using the comparative CT method. 18S ribosomal RNA (18S rRNA, assay Mm03928990_g1) was used as a reference gene to normalize gene expression data.

### 2.9. Histology

After routine formalin-fixed, paraffin-embedded specimen processing, 4 µm thick heart sections were prepared and stained with sirius red and hematoxylin–eosin (H&E) for the histological evaluation of cardiac fibrosis and morphology, respectively. For the quantitative analysis of sirius red staining, 8–10 random areas of the LV were selected from 5–6 different sections in each group using an Olympus BX-53 microscope (Olympus, Center Valley, PA, USA). Images were collected at 20× and subsequently analyzed using the NIH software ImageJ (https://imagej.net/ij/, accessed on 11 October 2023), and sirius red coverage was expressed as a percentage of total imaged area. We performed histological assessment on samples from the 30-day timepoint to assess any changes due to chronic + binge ethanol.

### 2.10. Statistical Analysis

Student’s *t* test or a one-way ANOVA was used to assess significant differences in data collected, as appropriate. Findings were expressed as mean and standard error of the mean, with the exception of weights, which were expressed as mean and standard deviation. Statistical analysis and presentation were completed using GraphPad Prism 9 (GraphPad Software, San Diego, CA, USA). *p* < 0.05 was considered significant. 

## 3. Results

### 3.1. Changes in Heart Weights

Body weight increased in control mice from day 30 to day 60 (Figure 2A: 30d Cntl: 30.2 ± 2.9 g compared to 60d Cntl: 40.9 ± 8.1, *p* < 0.01). At day 30, the ethanol mice (30d EtOH) heart weights were not significantly different from those of the control mice (30d Cntl); this was also true at day 60 (60d Cntl) (Figure 2B: 30d EtOH: 6.0 ± 0.1 mg/mm compared to 30d Cntl: 6.6 ± 0.1 mg/mm, and 30d/30d Abst: 7.1 ± 0.1 mg/mm compared to 60d Cntl: 7.7 ± 0.3 mg/mm). An increase in heart weight is expected during normal growth, and this was reflected in total heart weight comparisons from day 30 to day 60 in the control groups. LV weight showed a similar pattern (Figure 2C). There was no significant difference in 30d EtOH or 30d/30d Abst compared to both 30d and 60d control mice (30d EtOH: 4.0 ± 0.1 mg/mm compared to 30d Cntl: 4.5 ± 0.2 mg/mm; 30d/30d Abst: 5.0 ± 0.2 mg/mm compared to 60d Cntl: 5.3 ± 0.6 mg/mm). LV weights increased in both groups from day 30 to day 60. Right ventricular weight did not follow this pattern, and at no point did RV weights significantly differ except when comparing 30-day ethanol mice to 60-day control mice. Tibial lengths were not significantly different, except when comparing 30d EtOH mice to 30d/30d Abst mice; mice that had undergone abstinence had an increased tibial length (30d EtOH: 17.5 ± 0.2 mm compared to 30d/30d Abst: 18.2 ± 0.4 mm, *p* < 0.01; 30d Ctl 17.7 ± 0.5 mm; 60d Ctl 17.8 ± 0.3 mm). 

### 3.2. Echocardiography Data from Days 30 and 60

At 30 days, the LV anterior wall (LVAW) in both systole (s) and diastole (d) was not significantly different in 30d EtOH compared to 30d Cntl (Figure 3: in diastole, 30d EtOH: 0.75 ± 0.03 mm compared to 30d Cntl: 0.77 ± 0.09 mm, and in systole, 30d EtOH: 0.98 ± 0.04 mm compared to 30d Cntl: 0.89 ± 0.21 mm); the same was true with the posterior wall (LVPW) (in diastole, 30d EtOH: 0.65 ± 0.02 compared to 30d Cntl: 0.71 ± 0.03 mm, and in systole, 30d EtOH: 0.78 ± 0.03 mm compared to 30d Cntl: 0.84 ± 0.04 mm). This suggests that while our protocol induces significant cardiac dysfunction, it does not lead to significant structural changes in the LV wall. However, at 60 days, after mice had been abstained from alcohol, LVAW in both systole and diastole was significantly larger compared to 60d Cntl mice (Figure 3E,F) (in diastole, 30d/30d Abst: 0.91 ± 0.04 mm compared to 60d Cntl: 0.75 ± 0.01 mm, *p* < 0.01, and in systole, 30d/30d Abst: 1.15 ± 0.04 compared to 30d Cntl: 0.97 ± 0.02 mm, *p* < 0.01). The increased size of the LV dimensions could indicate compensatory mechanisms leading to cardiac hypertrophy. LVPW was not significantly different in systole, but there was a trend (*p* = 0.08) that LVPW in diastole was decreased in 30d/30d Abst mice. 

### 3.3. LV Catheterization Data from Days 30 and 60

Representative LV pressure-volumeume loops are shown in Figure 4 with group-averaged parameters in Figure 5. Both stroke work and dP/dt max are indicators of systolic function, while dP/dt min and Tau (the time constant of isovolumetric relaxation) are indicators of diastolic function. The 30d EtOH mice showed significant systolic and diastolic impairment compared to the 30d Cntl mice. Both systolic measures of stroke work and dP/dt max were decreased in the 30d EtOH mice (Figure 5A,B) (stroke work, 30d EtOH: 1119 ± 89 mmHg*µL compared to 30d Cntl: 1622 ± 128 mmHg*µL, *p* < 0.01) (dP/dt max, 30d EtOH: 8054 ± 665 mmHg/s compared to 30d Cntl: 11,188 ± 724 mmHg/s, *p* < 0.01). Ethanol mice had significantly reduced dP/dt min and a trend of decreased Tau (*p* = 0.0512) as compared to the 30-day control mice (Figure 5C,D) (dP/dt min, 30d EtOH: −7711 ± 561 mmHg/s compared to 30d Cntl: 10,147 ± 448 mmHg/s, *p* < 0.01) (Tau, 30d EtOH: 6.1 ± 0.3 ms compared with 30d Cntl: 5.2 ± 0.2 ms, *p* = 0.0512). Taken together, these findings illustrate that at 30 days, mice on ethanol had a significant reduction in both systolic and diastolic function. This contrasts with what was found at 60 days. The 30d/30d Abst mice had significantly increased stroke work and dP/dt max, though diastolic measures were not significantly different compared to the 60d Cntl mice (stroke work, 30d/30d Abst: 2032 ± 139.2 mmHg*µL compared to 60d Cntl: 1522 ± 171.7 mmHg*µL, *p* < 0.05) (dP/dt max, 30d/30d Abst: 11,967 ± 449.2 mmHg/s compared to 60d Cntl: 9787 ± 670 mmHg/s). Heart rates did not differ between the 30d EtOH mice and the 30d Cntl mice at catheterization (heart rate, 30d EtOH: 565 ± 12 bpm compared to 30d Cntl mice: 576 ± 5 bpm); this was also true at the 60-day timepoint (heart rate, 30d/30d Abst: 624 ± 20 compared to 60d Cntl: 591 ± 19 bpm). 

### 3.4. Load-Independent Measures

For these measures, the pattern of cardiac dysfunction in 30d EtOH mice was similar to that found in the load-dependent measures above. These mice had reduced E(es) and PRSW at 30 days (Figure 6A,B) (E(es), 30d EtOH: 6.9 ± 0.6 mmHg/µL compared to 30d Cntl: 12.4 ± 0.8 mmHg/µL, *p* < 0.0001) (PRSW, 30d EtOH: 78.6 ± 6.0 mmHg compared to 30d Cntl: 97.6 ± 3.3 mmHg, *p* < 0.05). At 60 days, the difference between 30d/30d Abst mice and 60d Cntl mice was not significant.

### 3.5. Cardiac Changes in Control Mice from 30 to 60 Days

We used these echocardiographic and catheterization measures to assess the changes in cardiac morphology and function in mice from 30 days to 60 days, as well (Figure 7 and Figure 8). In control mice, there were no significant changes in LV wall thickness or load-dependent measures of cardiac function, though there was a significant decrease in PRSW at 60 days (Figure 7) (PRSW, 60d Cntl: 83.5 ± 4.8 mmHg compared with 30d Cntl: 97.6 ± 3.3 mmHg). This implies that across most of the assessments, these control mice did not change significantly between 30 and 60 days on the liquid diet. 

### 3.6. Thirty Days of Abstinence Restores Both Diastolic and Systolic Function in Mice with ACM

Compared to 30d EtOH mice, LV dimensions and cardiac function significantly changed in 30d/30d Abst mice (Figure 8). LVAW increased in size, evident in both diastole and systole (LVAW;d, 30d EtOH: 0.75 ± 0.03 mm compared to 30d/30d Abst: 0.91 ± 0.04 mm, *p* < 0.01) (LVAW;s, 30d EtOH: 0.97 ± 0.04 mm compared to 30d/30d Abst: 1.15 ± 0.04 mm, *p* < 0.01) (Figure 8A,B). LVPW during systole was significantly increased, but this was not significant in diastole (Figure 8C,D). Catheterization measurements of systolic function, stroke work, and dP/dt max increased from 30 to 60 days, which indicates a significant increase in cardiac contractility (stroke work, 30d EtOH: 1119 ± 89 mmHg*µL compared to 30d/30d Abst: 2032 ± 139 mmHg*µL, *p* < 0.0001) (dP/dt max, 30d EtOH: 8054 ± 664 mmHg/s compared to 30d/30d Abst: 11,967 ± 449 mmHg/s, *p* < 0.01). The diastolic measure of dP/dt min increased in magnitude, suggesting an increase in the heart’s ability to relax (dP/dt min, 30d EtOH: −7711 ± 561 mmHg/s compared to 30d/30d Abst: −10,202 ± 594, *p* < 0.05) (Figure 8G), though Tau did not significantly change (Figure 8H). E(es) increased from 30 to 60 days (E(es), 30d EtOH: 6.9 ± 0.6 mmHg/µL compared to 30d/30d Abst: 10.6 ± 1.6 mmHg/µL, *p* < 0.05) (Figure 8I); PRSW did not significantly change (Figure 8J).

### 3.7. Reversal of Proinflammatory and Profibrotic Markers after Abstinence

Interleukin-6 (IL6) mRNA was significantly decreased in 30d/30d Abst mice compared to 30d EtOH (*p* < 0.05). There was a trending increase in IL6 mRNA in 30d EtOH mice compared to 30d Cntl mice (*p* = 0.07). Notably, there was not a significant difference between 30d Cntl mice and 60d Cntl mice (Figure 9A). Interleukin-1-beta (IL1β) mRNA was similarly decreased in 30d/30d Abst mice compared to 30d EtOH (*p* < 0.001). There was a significantly higher concentration of IL1β mRNA in 30d EtOH mice compared to both 30d Cntl (*p* < 0.01) and 60d Cntl (*p* < 0.01) (Figure 9B). Despite these reductions in proinflammatory cytokines, there was no significant change in IL10 mRNA after abstinence; there were no significant changes in IL10 mRNA in any of the groups (Figure 9C). Fibrotic biomarkers followed this pattern. La-ribonucleoprotein 6 (LARP6) mRNA was significantly higher in 30d EtOH mice compared to 30d Cntl mice (*p* < 0.05). Furthermore, after abstinence, there was a significant decrease in LARP6 mRNA in 30d/30d Abst mice compared to 30d EtOH mice (*p* < 0.05) (Figure 9D). COL1A1 mRNA also decreased in 30d/30d Abst mice compared to 30d EtOH mice (*p* < 0.05) (Figure 9E). There was a significant decrease in COL3A1 in the 30d EtOH mice (*p* < 0.0001) as compared to the 30d Cntl and 60d Cntl mice; COL3A1 expression was restored in the 30d/30d Abst group (Figure 9F). The ratio of COL1A1 to COL3A1 was increased in 30d EtOH mice compared to 30 Cntl mice (*p* < 0.001), and it was significantly reduced after abstinence in the 30d/30d Abst group (*p* < 0.001) (Figure 9F). 

### 3.8. Histological Collagen Staining Illustrated no Significant Differences in LV Collagen after 30 Days of Chronic + Binge Alcohol

Using picrosirius red staining, we performed histological assessment of LV sections from the 30d EtOH and the 30d Cntl groups. There were no statistically significant changes in collagen deposition in ethanol-exposed mice in comparison to control mice (percent area of fibrosis, 30d EtOH: 0.76 ± 0.01 compared to 30d Cntl: 0.66 ± 0.01, *p* = 0.46; Figure 10). This finding indicates a lack of overt fibrotic changes, despite an upregulation in COL1A1 mRNA and a decrease in diastolic function at 30 days in the alcohol-fed mice.

## 4. Discussion

As one of the most widely used drugs, alcohol can be consumed excessively and lead to AUD, which can ultimately cause death due to both primary and secondary pathological challenges. While many of these challenges involve hepatic and neural health, the heart is also adversely affected by excessive alcohol consumption, and the effects and magnitude of these effects are dependent on the dose and frequency of alcohol use, as well as individual risk factors such as genetics [26]. While some studies have shown that there may be slight benefits to drinking low amounts of alcohol [27], various pathological issues arise at different levels of consumption. Less than 20 drinks a week is associated with a decreased risk for both ischemic and hemorrhagic stroke [14], but consuming as little as 1.2 drinks a day significantly increases the risk of atrial fibrillation [9]. Over time, heavy alcohol use can lead to ACM, which has a poor prognosis. If patients do not completely abstain from alcohol use, there is a 50% chance of death within four years [3]. Despite this, human data and animal studies on the potential for restoration of cardiac function after developing ACM are limited. Before the diagnosis of ACM, there is an asymptomatic phase, but this asymptomatic phase can progress to systolic heart failure [28], from which there is no known cure. The underlying pathophysiology and the progression of the asymptomatic phase of ACM to heart failure are still being studied. Several researchers have investigated the direct cardiotoxic effects of alcohol and its metabolites [19], but the advancement to cardiac dysfunction has been difficult to describe, as the timeline and characteristics of the asymptomatic phase vary individually and may not warrant scrutiny. Furthermore, the exact amount of alcohol needed to reach symptomatic dysfunction is unknown. Diastolic dysfunction is present in at least 30% of patients with chronic alcohol consumption [29], and ventricular dilation can be the result of cardiotoxicity and of heart failure. Dilation leads to an increased end-diastolic and end-systolic volume, which impairs both diastole and systole; progressive dilation can also lead to morphological pathologies such as mitral and tricuspid regurgitation [30]. However, the order of events leading to ACM is unknown, and some clinical studies have even shown that depressed systolic function is present in asymptomatic patients with AUD [31]. A recent paper by Mirijello et al. discusses the uncertainty surrounding the progression of ACM to a symptomatic phase, characterized by systolic failure, dilation, and fibrosis [24].

Symptoms such as a decreased ejection fraction, dilation of the LV, and impairment of contractility [32] can be managed pharmacologically, but abstinence is the most recommended method of halting disease advancement. Our protocol of alcohol administration in mice resulted in significant diastolic and systolic dysfunction over the course of 30 days, and variations of this model have been used to study myocardial oxidative stress, mitochondrial stress, and cardiac dysfunction and steatosis [33]. To our knowledge, no studies have characterized functional recovery from ACM in a preclinical animal model, though in 1997, a clinical paper studying nine patients showed that remarkable increases in ejection fraction were possible with abstinence [34]. Therefore, our primary research goal was to investigate the effects of abstinence from alcohol on cardiac morphology and function in mice with established ACM. Our hypothesis was that abstinence would lead to improvement in cardiac function.

Our findings indicate that full functional recovery from alcohol-induced cardiac dysfunction is possible given a period of abstinence. Heart weights and echocardiography showed that significant cardiac dysfunction was present without overt morphological changes. Our chronic + binge model of alcohol consumption produced both systolic and diastolic dysfunction, evidenced by LV catheterization and pressure-volume loop analyses. Notably, the systolic dysfunction was apparent in load-independent measurements, indicating a reduction in cardiac contractility. After a period of abstinence, these mice made a remarkable recovery. LV catheterization showed that there was a recovery of diastolic function, as well as load-dependent and -independent systolic function. Echocardiography indicated that LV dimensions increased after abstinence, but this primarily occurred in the anterior wall.

Body weight, heart weight, and tibial length provided important information on the recovery of the abstained mice. First, their hearts increased in weight, particularly the LV, and this increase in size brought them to the same heart weights as mice that were on the control diet. Abstinence restored cardiac function back to a control level, and in some measures, exceeded control levels. A recent study showed that cardiac growth can be compromised by alcohol consumption in adolescent Wistar rats [35], which is consistent with our findings, though our mice were already in early adulthood. It is possible that alcohol consumption delayed the growth of their hearts, which rebounded after abstinence. There were no significant differences in body weight except for between the 30- and 60-day control mice and between 30-day ethanol mice and 60-day control mice. It is reasonable to expect these changes, as mice around this age should gain weight from 30 to 60 days. 

The heart can hypertrophy in different patterns to adapt to increased stress and demand. Our laboratory has previously used surgical approaches to induce pressure and volume overload and studied the resulting cardiac responses and pathological states [36,37,38]. We found that while alcohol can activate cardiac fibroblasts (and cause differentiation to cardiac myofibroblasts), which is consistent with findings by Law and Carver [39], chronic ethanol exposure also reduces adaptive responses of the heart to both pressure and volume overload [37,38]. Furthermore, alcohol causes accelerated eccentric remodeling in response to chronic volume overload [38]. Our model of chronic + binge alcohol consumption produced diastolic dysfunction as determined by LV catheterization, but echocardiography did not show a significant decrease or increase in LV dimensions at the 30-day timepoint. What was found, however, was that abstained mice exhibited increased LVAW thickness in both systole and diastole compared to control mice. Both the LVAW and the LVPW increased in size during abstinence compared to their size at 30 days. The increase in LVAW and LVPW can potentially be explained by abstinence removing the impairment that alcohol had on cardiac growth. Pressure-volume loop analysis is the gold standard for measuring in vivo cardiac function, as it is highly sensitive and yields an abundance of data regarding both load-dependent and -independent ventricular function. Stroke work is crucial in the context of systolic function and is the work needed to eject blood from the LV through the aortic valve. This measure describes mechanical efficiency. Stroke work was significantly lower in ethanol-fed mice versus the control mice, indicating that alcohol reduced contractility and systolic function. This effect of alcohol was completely reversed after the 30-day period of abstinence. 

dP/dt is the derivative of pressure (P) with respect to time (t); this is a measurement of the rate of change in pressure. dP/dt max refers to the maximum rate of pressure generated, while dP/dt min refers to the maximum rate of pressure decrease. The maximum pressure generated in a cardiac cycle occurs part way through the ejection phase, as the heart reaches its maximal contraction. Thus, dP/dt max can be thought of as a measurement of the rate at which peak contraction occurs during systole. The lowest pressure generated by the LV occurs at the end of isovolumetric relaxation; the chamber creates negative pressure by expanding while both the mitral and aortic valves are closed. Thus, dP/dt min is a measurement of the rate at which peak relaxation occurs during diastole. At 30 days, dP/dt max was significantly less in mice that had been consuming alcohol; this reflects a reduction in the ability of the heart to generate the necessary force to pump blood effectively. dP/dt min was also significantly reduced in these mice and reflects the reduced ability of the heart to relax efficiently. Both dP/dt min and max completely recovered after 30 days of abstinence. 

Tau, another index of diastolic function, is the time constant of isovolumetric relaxation; it characterizes the exponential decline in pressure as the ventricles relax and blood starts to flow back into the ventricles from the atria. There was a trend of Tau increasing in the mice that had consumed ethanol for 30 days (*p* = 0.051). A larger Tau indicates an increased amount of time needed to fully relax during diastole. This increase in Tau was completely reversed by abstinence.

Inferior vena cava occlusion during pressure–volume catheterization is used to assess load-independent measures of contractility, as occlusion decreases the volume of preload and measurements can be assessed over multiple volumes. The end-systolic pressure-volume relationship is linear, and as preload decreases, the end-systolic pressure and the end-systolic volume will also decrease. The slope of this line, E(es), or the ventricular elastance, is a load-independent measurement of cardiac contractility. Typically, a steeper slope indicates more forceful ventricular contraction. At thirty days, mice consuming the ethanol diet had a significantly decreased E(es), illustrating an intrinsic reduction in cardiac contractility, not dependent on Frank–Starling forces. This dysfunction was completely reversed after abstinence. Preload-recruitable stroke work (PRSW) uses inferior vena cava occlusion to plot stroke work against end-diastolic volume and is another measure of intrinsic contractility; a steeper slope indicates greater contractility. PRSW was significantly lower in mice on the ethanol diet at 30 days compared to control mice at 30 days, which, similarly to E(es), indicates that these mice had decreased systolic function, independent of preload. This decrease in contractility was also completely reversed with abstinence, as the 30d/30d Abst mice did not have significantly different PRSW values compared to control mice at 60 days. After abstinence, we observed a reduction in both inflammatory and fibrotic biomarkers in the LV, evidenced by qPCR. Chronic ethanol exposure in C57BL6/J mice increases pro-inflammatory cytokines in several tissues, including the heart [40], and clinically, IL6 and IL1β are known to be elevated with alcohol abuse [41]. We observed an increase in these biomarkers after 30 days of ethanol exposure. After abstinence, we observed a reduction in IL6 and IL1β compared to the levels of these biomarkers after 30 days of alcohol exposure. This decrease brought the 30d/30d Abst mice to control levels of IL6 and IL1β, as there was no significant difference between 30d/30d Abst mice and 60d Cntl mice (as well as 30d Cntl mice). This reduction in inflammation after abstinence has also recently been found in the liver, in C57BL6/J mice fed the Leiber-DeCarli liquid diet with ethanol for 28 days [42]. Taken together, these data suggest that while chronic alcohol does lead to inflammation, resolution is possible with the cessation of alcohol intake. Chronic alcohol exposure increases collagen expression in some animal models [43], and cardiac fibrosis is a hallmark of clinical ACM [21]. Collagen I and III serve as structural support in the heart, but collagen I is more stiff, while collagen III is more elastic [44]. We found significant diastolic dysfunction after 30 days of alcohol exposure. We also observed an increase in COL1A1 and a decrease in COL3A1 expression, which could indicate a stiffer heart. This ratio of COL1A1 to COL3A1 returned to normal after abstinence, which was consistent with our functional findings of restored diastolic function in those animals. We also found an increase in LARP6 in the 30d EtOH mice; LARP6 is a positive regulator of collagen expression [45], and inhibition of LARP6 decreases hepatic fibrosis resulting from alcohol exposure [46]. In 30d EtOH mice, LARP6 was increased relative to the control, but after abstinence, mRNA abundance decreased to control levels. While cardiac fibrosis is a key finding in clinical ACM, preclinical models do not accurately represent these changes in cardiac morphology. Despite finding a reduction in diastolic function and increased expression of COL1A1 after 30 days of alcohol, we did not find evidence of LV fibrosis in histological sections.

Alcohol and its effects have been studied in various animal models and ethanol intake paradigms, all with their own strengths and weaknesses, but fewer studies focus on the cardiac effects of alcohol. A hallmark of clinical ACM is significant cardiac fibrosis, which can arise from various pathogenic processes including microvascular ischemia, increased inflammation, neurohormonal activation, wall stress, and direct damage [47]. Fibrosis is often a secondary concern in clinical ACM, as patients typically present with symptoms of heart failure and systolic dysfunction, which arise after fibrosis and diastolic dysfunction. Although fibrosis occurs prior to overt systolic failure, cardiac morphology, dimensions, and apparent fibrosis can aid in the staging of a patient’s risk. Few preclinical studies using alcohol administration have been able to mimic the extensive cardiac fibrosis present in clinical ACM, though one genetic mouse line with knocked-out metallothionine expression exhibited cardiac fibrosis with heavy alcohol exposure [48]. Alcohol and its metabolites (acetaldehyde and reactive oxygen species (ROS)) are cardiotoxic and have direct toxic effects on the myocardium. Cardiomyocyte dysfunction could also occur through disruption of mitochondrial function by alcohol [33,49] or through activation of neurohormonal signaling, such as the renin–angiotensin–aldosterone system [5,50].

While the findings of this study provide valuable insights into the topic at hand, it is essential to acknowledge limitations that warrant consideration in interpreting the results. While we observed a functional recovery after abstinence in a preclinical model, humans consume alcohol for years, at varying amounts, and the point at which irreversible damage occurs is unknown [51]. No animal models perfectly mimic clinical ACM, and it is particularly difficult to reproduce the extensive fibrosis typically present. The chronic + binge ACM mouse model does not lead to overt fibrosis but does produce robust and consistent cardiomyopathy with both diastolic and systolic functional depression as seen clinically. Another limitation of our study is the absence of a female cohort; these studies are currently underway, but further investigation is needed. While clinical AUDs affect males disproportionately more than females [52], females, in several pathological measures, have worse health outcomes related to alcohol-induced damage [53]. Given the limited existing data and knowledge on reversal of cardiac damage with abstinence, we chose to initially concentrate on a male cohort to establish a foundational understanding before potentially expanding our investigation to include female animals. It is not known if the chronic + binge models will exhibit sex differences in cardiac dysfunction or recovery after abstinence.

## 5. Conclusions

In this study, we found that alcohol-induced cardiac dysfunction in mice can be completely reversed with abstinence; this was illustrated by echocardiography and LV catheterization. In clinical ACM, while fibrosis is largely irreversible, there is some evidence that cardiac function can return to normal after the cessation of drinking. However, there are limited studies that provide clear data that support this proposal. Here, we used a preclinical mouse model of ACM to investigate potential reversal of cardiac dysfunction and found that complete recovery of both systolic and diastolic function occurred. Furthermore, abstinence also resolved elevation in both fibrotic and inflammatory biomarkers. Moving forward, it is necessary to investigate potential molecular explanations of these reparative processes, with the eventual goal of finding an intervention that can prevent or reverse the damage alcohol causes to the heart or treatments that can hasten natural recovery. 

## Figures and Tables

**Figure 1 cells-12-02783-f001:**
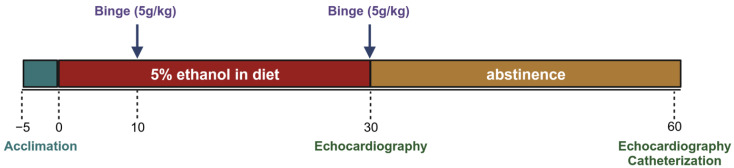
Experimental timeline: Acclimation to liquid diet was allowed for 5 days, before animals were switched to ethanol diet. At days 10 and 30, animals were gavaged with 5 g/kg ethanol or isocaloric 9 g/kg maltose–dextrin. Echocardiography was performed at least 12 h after gavage. Terminal LV catheterization was performed the day after echocardiography was completed. There were 4 groups in total: mice receiving alcohol for 30 days with 2 binge doses (30d EtOH), mice on control diet for 30 days (30d Cntl), mice that after 30 days of alcohol exposure (and 2 binges) were transitioned to control diet (30d/30d Abst), and mice that received the control diet for 60 days (60d Cntl).

**Figure 2 cells-12-02783-f002:**
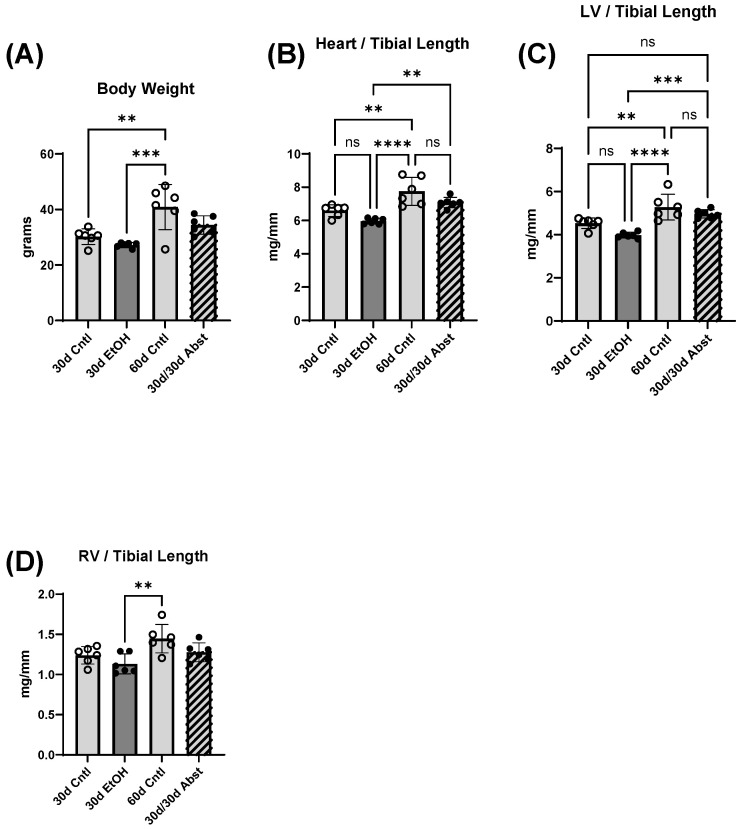
Heart and body weights (**A**) Body weights. (**B**–**D**) Hearts weights were normalized using tibial length. These data were analyzed using one way ANOVA (LV: left ventricle; RV: right ventricle; ns = not significant, ** *p* < 0.01, *** *p* < 0.001, and **** *p* < 0.0001).

**Figure 3 cells-12-02783-f003:**
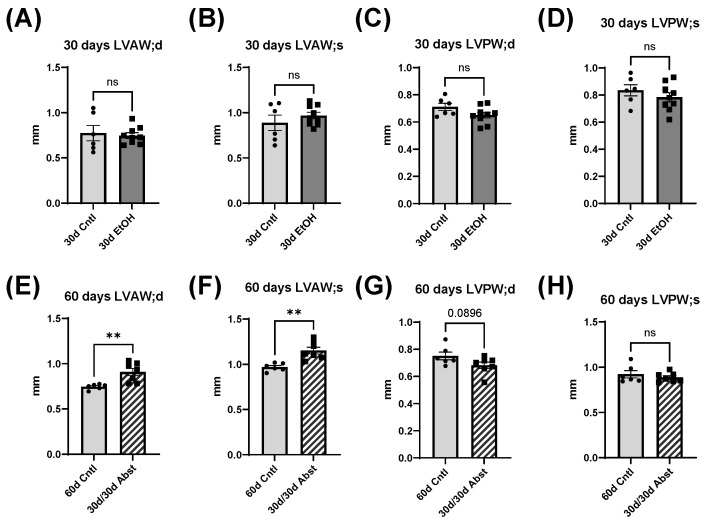
Echocardiography data at 30 and 60 days. (**A**–**D**) There were no significant changes in LV wall dimensions at 30 days. (**E**–**H**) LV anterior wall (LVAW) in both systole (s) and diastole (d) increased in 30d/30d Abst mice compared to 60d Cntl mice (** *p* < 0.01) with no significant difference in LV posterior wall (LVPW). Data were analyzed using Student’s *t* test (ns = not significant).

**Figure 4 cells-12-02783-f004:**
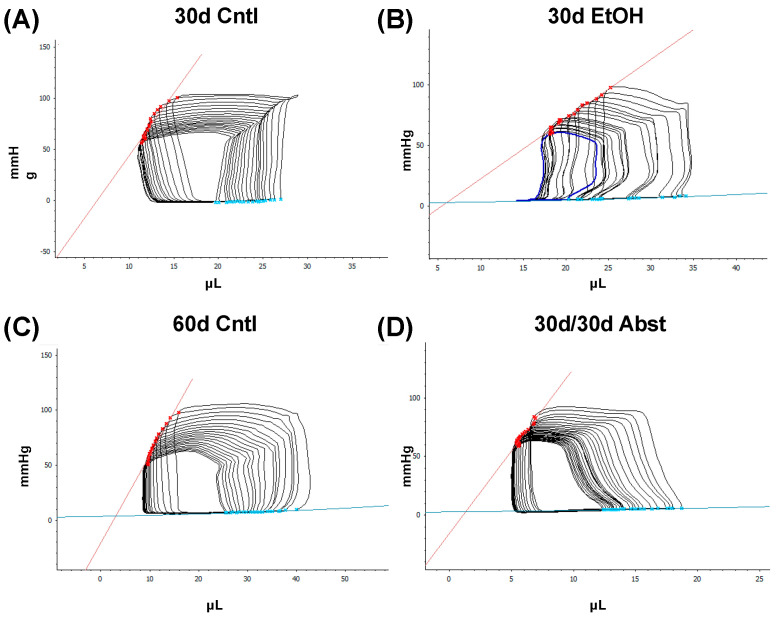
Representative LV pressure-volume loops for each group. Red lines indicate end-systolic pressure-volume relationship (ESPVR), while blue lines indicate end-diastolic pressure-volume relationship (EDPVR). Steady-state pressure-volume loops for: (**A**) 30d Cntl mice, (**B**) 30d EtOH mice, (**C**) 60d Cntl mice, and (**D**) 30d/30d Abst mice.

**Figure 5 cells-12-02783-f005:**
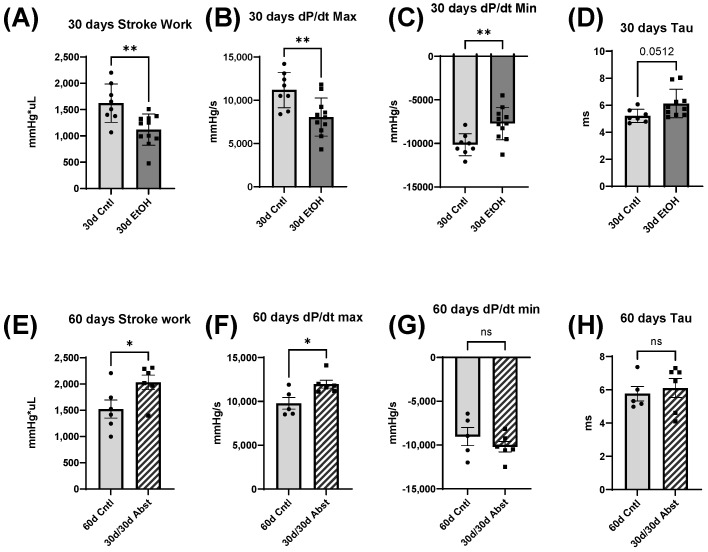
LV catheterization data at 30 and 60 days. (**A**–**D**) 30d EtOH mice had decreased systolic and diastolic function compared to 30d Cntl mice (** *p* < 0.01). (**D**) There was a trend for reduction in time relaxation constant, Tau (*p* < 0.051). (**E**–**H**) At 60 days, 30d/30d Abst mice exhibited increased systolic function, and diastolic function increased to match 60d Cntl diastolic function. (**E**) Stroke work and (**F**) dP/dt max were significantly increased compared to 60d Cntl mice (* *p* < 0.05). (**G**–**H**) Both dP/dt min and Tau were not significantly different versus 60d Cntl mice. Data were analyzed using Student’s *t* test (ns = not significant).

**Figure 6 cells-12-02783-f006:**
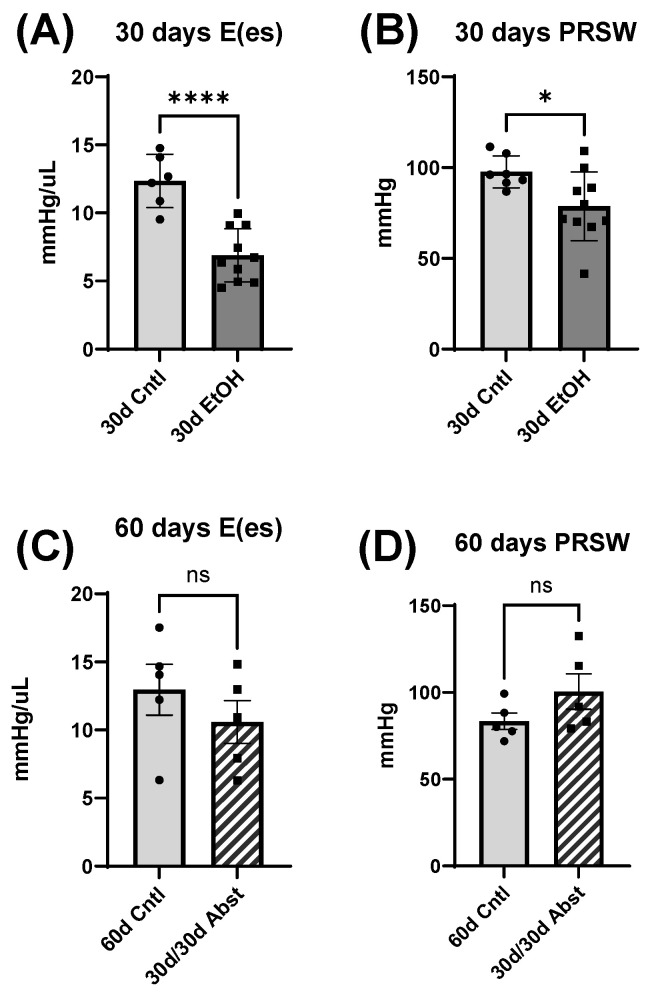
Load-independent measures of cardiac contractility. (**A**,**B**) At the 30-day timepoint, the slope of the end systolic pressure–volume relation, E(es), and preload-recruitable stroke work (PRSW) were significantly decreased in the 30d EtOh mice (* *p* < 0.05, **** *p* < 0.0001). (**C**,**D**) This reduction was abrogated by abstinence; both E(es) and PRSW in the 30d/30d Abst mice were not significantly different versus control mice at 60 days. Data were analyzed using Student’s *t* test (ns = not significant).

**Figure 7 cells-12-02783-f007:**
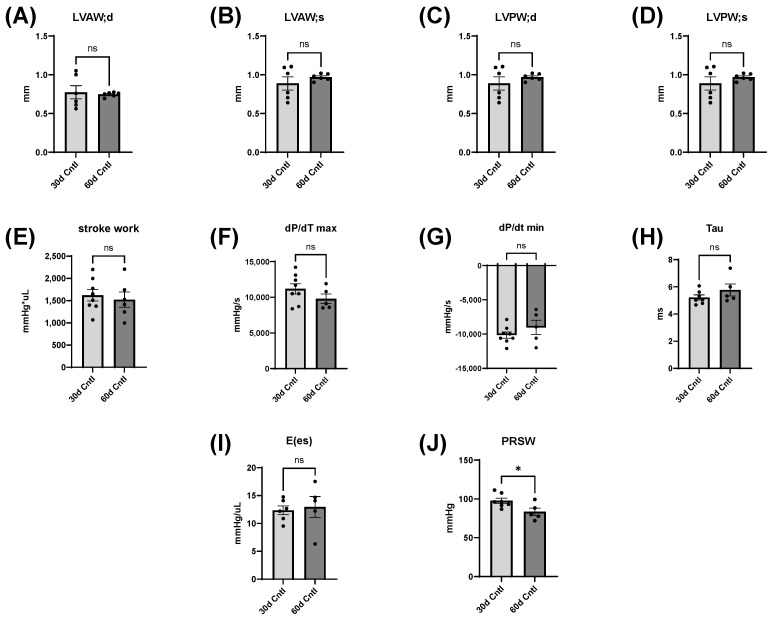
Echocardiography and LV catheterization data in control mice from day 30 to day 60. (**A**–**H**) There were no significant differences in LV wall dimensions or load-dependent measures of contractility. (**I**) E(es) was also not significantly changed. (**J**) PRSW significantly decreased at day 60 (* *p* < 0.05). Data were analyzed using Student’s *t* test (ns = not sinificant).

**Figure 8 cells-12-02783-f008:**
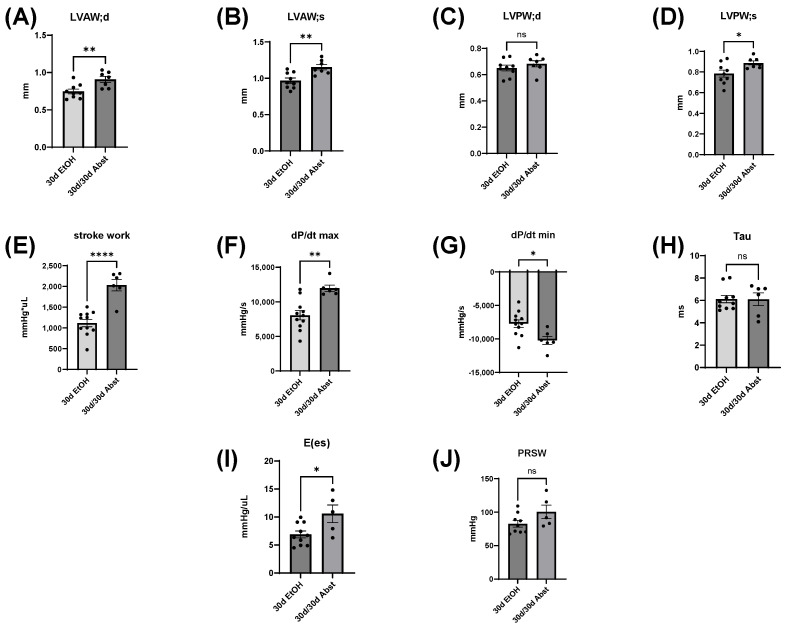
Echocardiography and LV catheterization data in 30d EtOH mice compared to 30d/30d Abst mice. (**A**,**B**) LVAW size increased in both systole and diastole with abstinence. (**C**,**D**) While LVPW in diastole did not significantly change, LVPW in systole increased. (**E**,**F**) Both systolic measures of stroke work and dP/dt significantly increased in 30/30d Abst mice. (**G**) The diastolic measure dP/dt min also significantly increased. (**H**) Tau was not significantly changed. (**I**,**J**) E(es) significantly increased with abstinence, while PRSW did not. Data were analyzed using Student’s *t* test (* *p* < 0.05, ** *p* < 0.01, **** *p* < 0.0001; ns = not significant).

**Figure 9 cells-12-02783-f009:**
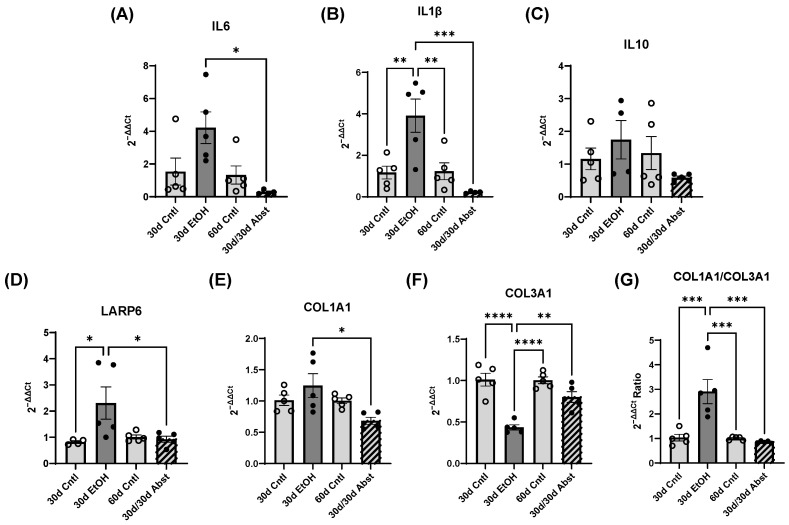
mRNA assessment using qPCR. (**A**–**C**) Relative amounts of inflammatory cytokines. (**A**) IL-6 was significantly decreased after abstinence (* *p* < 0.05). (**B**) IL1β was significantly increased after 30 days of alcohol exposure compared to control (** *p* < 0.01) and was significantly decreased after abstinence (*** *p* < 0.001). (**C**) IL10 was not significantly changed by alcohol exposure or abstinence. (**D**–**G**) Relative amounts of fibrotic markers. (**D**) LARP6 was significantly increased after 30 days of alcohol exposure (* *p* < 0.05) and returned to control levels with abstinence. (**E**) COL1A1 was significantly decreased after abstinence relative to 30d EtOH. (**F**) COL3A1 was decreased with 30 days of alcohol exposure (**** *p* < 0.0001) and returned to near control levels after abstinence. (**G**) The COL1A1/COL3A1 ratio was increased after 30 days of alcohol exposure (*** *p* < 0.001) and restored to control levels by abstinence (*** *p* < 0.001). Data were analyzed using a one-way ANOVA, and post hoc analysis was performed using Tukey’s multiple comparisons test.

**Figure 10 cells-12-02783-f010:**
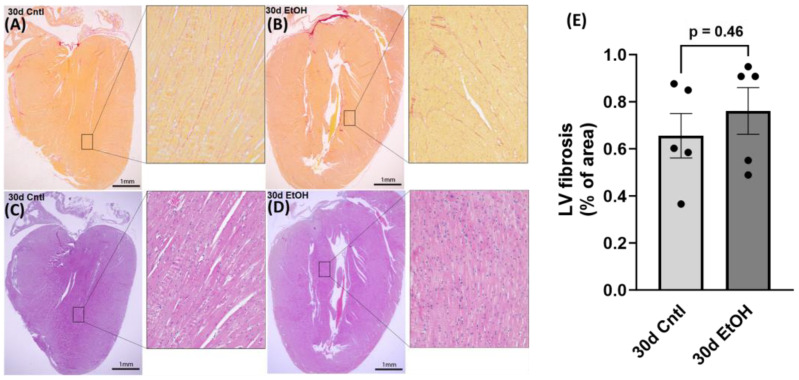
(**A**,**B**) Representative images of picrosirius red staining of collagen (2× macro; 20× inset). (**C**,**D**) Representative images of hematoxylin and eosin staining. (**E**) Quantification of picrosirius red staining found no significant changes in collagen deposition after 30 days of alcohol. Macro images were edited using Adobe Photoshop to remove cutting artifact and improve color. The 20× images of LV that were used for collagen quantification were processed raw and not modified.

## Data Availability

Raw data are available from the corresponding author upon request.

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
