# Peer review of "Abstinence Restores Cardiac Function in Mice with Established Alcohol-Induced Cardiomyopathy"

_cells, 2023, doi:10.3390/cells12242783_

Round 1
Reviewer 1 Report
Comments and Suggestions for Authors
This is an interesting work aiming to evaluate, in a mice model, the effect of alcohol on heart and the potential recovery of alcohol induced damage after abstinence.
Despite the paper sounds interesting, there are several flaws that limit its readability.
1) ABSTRACT: please remove references from abstract. Moreover, the abstract presentation should be improved in order to report results wimilarly to the main manuscript.
2) Introduction: the introduction section can be improved. In particular, the main problem (alcohol and heart) should be presented earlier instead of describing the effect of alcohol on the liver. Moreover, the effect of acute and chronic alcohol abuse should be differentiated.
Please include among references also: Fernández-Solà J. The Effects of Ethanol on the Heart: Alcoholic Cardiomyopathy. Nutrients. 2020 Feb 22;12(2):572. doi: 10.3390/nu12020572. PMID: 32098364; PMCID: PMC7071520.
3) Introduction, lines 55-57: This sentence should be rewritten in order to include an experimental paper describing the effect of alcohol abstinence on echocardiographic markers of alcohol abuse (Mirijello A, Sestito L, Lauria C, Tarli C, Vassallo GA, Antonelli M, d'Angelo C, Ferrulli A, Crea F, Cossari A, Leggio L, De Cosmo S, Gasbarrini A, Addolorato G. Echocardiographic markers of early alcoholic cardiomyopathy: Six-month longitudinal study in heavy drinking patients. Eur J Intern Med. 2022 Jul;101:76-85. doi: 10.1016/j.ejim.2022.04.005. Epub 2022 Apr 10. PMID: 35418346; PMCID: PMC9233051).
4) Introduction, line 59: Thomas et al. The manuscript has only 1 author. Please refer as Thomas.
5) Introduction, lines 75-78: Results are supposed to be presented in the results section instead of intro section.
6): Methods section: the design of the study should be described more clearly, in a separate paragraph/subparagraph, even including to figure 1.
7) results: In the Results section you should limit to describe results. Please do not report any methodology, as in lines 172-176; 179-181; 200-202; etc. This section should be rewritten as well as methods section, accordingly.
8) discussion: The exact progression of ACM from normal heart to 4 chambers dilation is still matter of debate. Please expand these uncertainties, even referring to a recent review paper (Mirijello A, Tarli C, Vassallo GA, Sestito L, Antonelli M, d'Angelo C, Ferrulli A, De Cosmo S, Gasbarrini A, Addolorato G. Alcoholic cardiomyopathy: What is known and what is not known. Eur J Intern Med. 2017 Sep;43:1-5. doi: 10.1016/j.ejim.2017.06.014. Epub 2017 Jun 21. PMID: 28647343).
9) Limitations: this paper is focused on assessing the reversibility of ACM after 1 month of exposure to alcohol. However, one of the main questions open in clinical practice is to identify the no-return point, in other words, the cut-off for reversibility or not (please see and cite : Fernández Solà J. Reversibility of Alcohol Dilated Cardiomyopathy. Rev Esp Cardiol (Engl Ed). 2018 Aug;71(8):603-605. English, Spanish. doi: 10.1016/j.rec.2018.01.016. Epub 2018 Apr 10. PMID: 29653776). The design of this study can't evaluate these aspects and it should be underlined among limitations.
Please revise English/typo errors.
Comments on the Quality of English LanguageEnglish Language requires minor check.
Author Response
We thank the reviewers for their thoughtful suggestions towards improving our manuscript “Abstinence restores cardiac function in mice with established alcohol-induced cardiomyopathy”. Responses are addressed below.
Reviewer 1 highlighted various grammatical issues in the introduction that have been resolved. References were removed from abstract. The 2nd paragraph has been replaced with a paragraph addressing the relationship between alcohol and the heart. This is followed by a paragraph on the acute effects of alcohol. The reference to Thomes was corrected on line 70. Lines 77-80 now include the suggested reference to the longitudinal study of clinical ACM using echocardiography to assess functional changes after abstinence. All results in the introduction were removed and placed in the results section.
Reviewer 1 also made suggestions for the methods section. An additional paragraph has been added to the methods section to clarify the experimental timeline, and the experimental design information has been removed from the results section.
Reviewer 1 included some suggestions for the discussion section as well. Lines 442-456 address the uncertainty regarding the progression of ACM. Lastly, the limitation of this study to address the point at which irreversible damage occurs has been added in lines 596-598.
Reviewer 2 Report
Comments and Suggestions for Authors
Reviewer report
Abstinence restores cardiac function in mice with established 2 alcohol-induced cardiomyopathy.
This is a well-described and explanatory study showing functional mechanisms by echocardiography and other techniques of alcohol-induced cardiomyopathy.
1. Study was performed in male mice what could be the difference if the study was performed in female mice will there be any gender variation?
2. In the echocardiography method section please mention the isoflurane concentration used.
3. Please mention the Vevo lab software version used eg. Vevo lab 5.7.1
4. There are a few more parameters that must have been estimated by the Vevo system other than Left ventricular wall thickness like Ejection fraction, Fractional shortening, Cardiac output, Volume for systole, Volume for diastole. Could authors elaborate on the effect of these parameters?
5. Authors could have performed color Doppler velocity measurements for mitral valve E/A ratio which is a measure of diastolic dysfunction.
6. There are no figure number legends and numbers for figures after figure 4 please add those.
Author Response
We thank the reviewers for their thoughtful suggestions towards improving our manuscript “Abstinence restores cardiac function in mice with established alcohol-induced cardiomyopathy”. Responses are addressed below.
Reviewer 2 asked about the possibility of variation between sexes, as our study did not present data including female mice. In human studies, it has been shown that while men drink more alcohol, have a greater prevalence of alcohol use disorders, and develop more alcohol-withdrawal symptoms compared to women; it has also been shown that women develop more severe pathological issues secondary to alcohol use. We are interested in exploring these sex-related differences and have run a preliminary cohort of age-matched female mice. We processed both echocardiographic and catheterization data but found no dysfunctional phenotype associated with the same protocol of alcohol exposure, suggesting that female C57BL6/J mice are protected. However, since we did not produce cardiac dysfunction in our female cohort, we did not continue to use females in the current study. In a future study, we intend to perform a more comprehensive analysis of potential sex differences and the role of ovarian hormones using our model of ACM.
Reviewer 2 also asked about specific details regarding our use of echocardiography in this study. The concentration of isoflurane has been included in the echocardiography method section (line 155). The Vevo lab software version used has also been included (line 159). While ultrasound echocardiography can provide estimates of cardiac function, specifically EF, FS, and CO, left ventricular pressure-volume catheterization is a more sensitive method of assessing ventricular function and provides both load-dependent and –independent indexes of contractility. However, the catheterization method does not provide measures of wall thickness, so we have included wall dimension data from ultrasound echocardiography. We did not study diastolic or systolic function using echocardiography using Doppler velocity, as the pressure-volume catheterization provides precise assessment of both.
Reviewer 2 also found an absence of figure number legends and numbers for figures after figure 4. Figure numbers and legends have been updated.
Reviewer 3 Report
Comments and Suggestions for Authors
The authors observed that abstinence rescues alcohol-induced cardiomiopathy in mice. The findings were demonstrated by performing gravimetric, echocardiographic, and hemodynamic analyses.
The work is interesting and extends previous evidence about the protective effects of alcohol consumption cessation. The analyses are also accurate.
I have some comments for the authors:
-Functional analyses should be confirmed by a morphological evaluation of the heart. I suggest to perform histological analyses in heart sections, in order to evaluate the extent of hypertrophy, fibrosis, as well as molecular markers of cardiac injury.
-It should be interesting to evaluate levels of apoptosis or necrosis, by western blot or Tunel assay
-It is well known that chronic alcohol consumption increases inflammation and oxidative stress. The authors should analyze the reduction of these mechanisms following abstinence
Author Response
We thank the reviewer for their thoughtful suggestions towards improving our manuscript “Abstinence restores cardiac function in mice with established alcohol-induced cardiomyopathy”. Responses are addressed below.
Reviewer 3 inquired about the use of morphological evaluation of the heart to confirm functional analyses, specifically suggesting histological analysis and assessment of molecular markers of cardiac injury. Functional analyses are now accompanied by histological assessments of cardiac tissue at 30 days with quantification of collagen staining in the left ventricle (Figs 9-10).
Reviewer 3 also suggested that we evaluate levels of apoptosis or necrosis, as well as markers of inflammation and oxidative stress. We performed qPCR for various inflammatory and fibrotic biomarkers which have been previously associated with alcohol intake (Fig 9). We found elevated IL6 and IL1b in the 30d EtOH mice, which was resolved in the 30d/30d Abst mice. We also found increased Larp6 and Col1a1 in 30d EtOH mice, and higher Col1a1/Col3a1 ratio, all of which were resolved in the abstinence group. Despite these increases in mRNA expression driven by EtOH, we found no significant differences in left ventricular collagen histologically (picrosirius red staining; Fig 10).
Round 2
Reviewer 1 Report
Comments and Suggestions for Authors
Authors have addressed all the raised comments.
Comments on the Quality of English LanguageEnglish Language is fine.
Reviewer 3 Report
Comments and Suggestions for Authors
The authors addressed all my comments, performing additional experiments. No further comments.